# Stable High-Capacity Elemental Sulfur Cathodes with Simple Process for Lithium Sulfur Batteries

**DOI:** 10.3390/molecules28124568

**Published:** 2023-06-06

**Authors:** Shunsuke Sawada, Hideki Yoshida, Shalom Luski, Elena Markevich, Gregory Salitra, Yuval Elias, Doron Aurbach

**Affiliations:** 1R&D Division, Nichia Corporation, Anan 774-8601, Japan; 2Department of Chemistry, Bar-Ilan University, Ramat Gan 5290002, Israel

**Keywords:** lithium sulfur battery, cathode, elemental sulfur, areal capacity, carbon paper, swelling

## Abstract

Lithium sulfur batteries are suitable for drones due to their high gravimetric energy density (2600 Wh/kg of sulfur). However, on the cathode side, high specific capacity with high sulfur loading (high areal capacity) is challenging due to the poor conductivity of sulfur. Shuttling of Li-sulfide species between the sulfur cathode and lithium anode also limits specific capacity. Sulfur-carbon composite active materials with encapsulated sulfur address both issues but require expensive processing and have low sulfur content with limited areal capacity. Proper encapsulation of sulfur in carbonaceous structures along with active additives in solution may largely mitigate shuttling, resulting in cells with improved energy density at relatively low cost. Here, composite current collectors, selected binders, and carbonaceous matrices impregnated with an active mass were used to award stable sulfur cathodes with high areal specific capacity. All three components are necessary to reach a high sulfur loading of 3.8 mg/cm^2^ with a specific/areal capacity of 805 mAh/g/2.2 mAh/cm^2^. Good adhesion between the carbon-coated Al foil current collectors and the composite sulfur impregnated carbon matrices is mandatory for stable electrodes. Swelling of the binders influenced cycling retention as electroconductivity dominated the cycling performance of the Li-S cells comprising cathodes with high sulfur loading. Composite electrodes based on carbonaceous matrices in which sulfur is impregnated at high specific loading and non-swelling binders that maintain the integrated structure of the composite electrodes are important for strong performance. This basic design can be mass produced and optimized to yield practical devices.

## 1. Introduction

Lithium sulfur (Li-S) batteries are expected to fulfill demands for high energy density and cost effectiveness due to high theoretical specific capacity (1674 mAh/g) and abundance of lithium sulfur in the Earth’s crust [1]. However, Li-S batteries cannot be considered as a commercial reality due to two main reasons: (1) poor cycling performance and safety risks related to the use of Li metal anodes (dendrite formation, highly reactive surface) and flammable electrolyte solutions [2] and (2) poor cycling capability of sulfur cathodes caused by self-discharge of intermediate polysulfide species (LiS*_n_*), the well-known shuttling effect [3].

Three approaches are used to prevent shuttling: (a) LiNO_3_ additive in DOL/DME enhances passivation on the Li metal anode through oxidation, which forms surface Li-S-O moieties [4]; (b) sulfur–carbon composite cathodes that accommodate sulfur inside carbon pores and suppress dissolution of reduction products in the electrolyte solution [3]. This method stabilizes Li-S cell performance; however, the sulfur–carbon composite structure limits the total amount of sulfur in the cathode, decreasing cell capacity; and (3) unique electrolyte solutions such as ionic liquids and additives prevent dissolution of LiS*_n_* species formed by sulfur reduction. Solid-state Li-S batteries with solid electrolytes [5] avoid shuttling but are not cost effective due to high resistance and slow operation.

Sulfur–carbon composites with a judicious use of reactive additives, which enhance the Li metal anode passivation, are the most realistic approach. The actual sulfur content in Li-S cells, compared to other necessary non-active components, is not sufficient to realize their theoretical energy density. Maintaining sufficient sulfur in composite cathodes is desired for high cell capacity and energy density. Battery manufacturers aim to use non-encapsulated elemental sulfur as the main cathode material to increase the total amount of sulfur in composite cathodes [6,7,8].

Elemental sulfur as an active mass has the potential to achieve high energy density and is cost-effective; however, achieving high sulfur content with high-rate capability composite electrodes is extremely challenging. Three major issues have been addressed: (1) insufficient electric conductivity of sulfur as an insulator was improved by utilizing porous current collectors that ensure good electronic integrity and high sulfur content in the pores [9]; (2) shuttling due to dissolution of sulfur in the electrolyte solution was mitigated by active additives that enhance the passivation of the lithium anodes [10,11]; (3) detachment of the binder from the current collector due to sulfur dissolution was largely avoided [12,13].

However, despite these efforts, high areal loading with high sulfur content remains a significant challenge; there are few reports on this issue [10]. In many cases, high electrode areal loading with low sulfur content or low areal loading with high sulfur content are used. Here, we focus on preparation of a high areal capacity cathode (high sulfur content and loading) with a long cycle life by using elemental sulfur as active material. In general, the capacity of such cathodes increases with the sulfur content, but when both active mass per square area and electrode loading are high, the specific capacity decreases drastically due to low utilization of the active mass [14].

Here, cathodes with high sulfur content and areal loading are developed with an effective 3D structure, aiming at 70% sulfur content in the electrode mass slurry [15]. Our focus is on several important engineering aspects of composite sulfur cathodes that determine their loading, durability, and effectiveness. It is important to emphasize that this and related work do not include demonstration of optimal systems with a minimal amount of unreactive materials and practically high specific capacity of electrodes and Li-S cells with high energy density and very prolonged cycle life. Here, ideas are presented on 3D structures of composite sulfur electrodes that may enable maximization of high specific areal loading of sulfur in practical Li-S batteries, which can be relatively easily applied in commercial production lines.

## 2. Results and Discussion

For cathodes based on elemental sulfur as active materials in rechargeable Li batteries, both the current collector and the binder must be optimized, as elemental sulfur is highly insulating, and its reduction to Li_2_S involves complex conversion reactions that form various Li*_m_*S*_n_* species. Polyethylene oxide (PEO), which can prevent shuttling by swelling, was used as a binder [12]. Furthermore, as PEO has relatively high resistance, polyvinyl alcohol (PVA) was mixed with it [16] using S/C composite cathode materials. As electrodes comprising elemental sulfur have more severe shuttling, we repeated the experiments with PEO/PVA binder. In recent studies on Li-S, the current collector is also important. The use of Al foil is relevant. However, the question is whether to use a glossy Al foil or to employ a composite and more sophisticated current collector. Here we propose carbon-coated Al foil as a current collector for composite sulfur cathodes. The carbon coating may require further optimization.

Two current collectors were studied at RT in Li-S half cells with S loading of 1.3 g/cm^2^ in 1 M LiTFSI in DOL/DME electrolyte solution between 2.5–1.0 V (vs. Li). Initial property of the cells were tested at 0.1 C rate (1 C = 1674 mAh/g) and then at 0.2 C. Figure 1 shows SEM images of the current collectors and the initial charge/discharge voltage profiles of the composite electrodes with different current collectors and binders. Carbon particles were confirmed on top of the carbon-coated current collector. Its thickness was less than 1 μm.

When glossy Al foils are used as current collectors, PEO binder adds high resistance to the composite electrodes. The addition of PVA improves the initial charge/discharge capacity; however the maximal specific capacity obtained with PEO/PVA binder is very low (420 mAh/g). Interestingly, while the same foils with PEO binder show extremely low initial capacity, a very high initial capacity is obtained with carbon-coated current collectors. These results suggest that PEO can alleviate the shuttling effect that reduces the specific capacity of sulfur electrodes, provided that the poor electronic conductivity at the current collector/active mass interface is enhanced. Hence, increasing electronic conductivity is important for composite electrodes based on elemental sulfur due to the intrinsic insulating nature of sulfur and its reduction products (Li*_m_*S*_n_* species, 0.5 < *n* < 4). It thus seems mandatory to use carbon-coated current collectors, which provide good interfacial electronic conductivity between the current collector and the active mass.

To confirm the qualitative difference between carbon-coated Al foil and glossy Al, the impedance of the Li-S cells with cathodes containing PEO binder was measured from 0.1 Hz to 1 MHz before cycling. The impedance is significantly lower (Figure 2) with carbon-coated foil. An additional conductive source (such as carbon-coated Al foil) is thus required for high initial capacity, especially with elemental sulfur as the active material.

EIS is often used in research works related to development of electrochemical power sources. The impedance response of electrochemical systems includes involvement of many processes that occur in parallel: adsorption of ions to surfaces, migration of ions through surface films, and charge transfer by ions and electrons through interfaces and bulk components (such as surface films, active masses, electrolyte systems). Hence, the time constants of the relevant processes are involved. Separation according to the impedance spectral features may be extremely difficult. This limitation of EIS is especially problematic in the study of composite battery electrodes such as the systems covered here. Hence, it is important to note that modelling impedance spectra of composite electrodes of the type used here may be meaningless due to the many possible time constants related to charge transfer in such electrodes. However, EIS provides qualitative insights through comparative impedance measurements. For instance, EIS measurements can indicate a dynamic situation, such as the involvement of side reactions through impedance increase during storage and/or cycling. It can also indicate the contribution of certain components in composite electrodes to their charge transport properties, as demonstrated by comparison of impedance spectra in Figure 2, which clearly shows the importance of the thin carbon coating on the Al current collectors used for sulfur cathodes.

Initial high discharge capacity was achieved using both the PEO binder and carbon-coated current collector. However, as described in the previous section, the sulfur loading was extremely low (only 1.3 mg/cm^2^). To achieve high sulfur loading 3D porous structured carbon paper (Toray H30, 80% porosity) was studied as part of the active mass. The use of a 3D structure is intended to provide better mechanical integrity and electrical conductivity throughout the composite active mass [9]. Porous carbon paper matrices can bestow conductivity to embedded elemental sulfur; however, their actual conductivity is much lower than metallic current collectors. Hence, this is not the best option to obtain both high sulfur loading and low resistance. Sulfur-impregnated 3D activated paper matrices were combined with carbon-coated Al foil (see Figure 3); a sulfur slurry was coated on the carbon paper, and the paper with sulfur was bound to the coated foil. The boundary between the two components was adhered by the PEO binder. A cross-sectional SEM image is shown in Figure 3b. The white area in the carbon paper is sulfur, which was confirmed to be vertically located in the entire carbon paper. It was also confirmed that the carbon paper was thick in relation to the amount of sulfur, even though the thinnest paper was selected. As it is difficult to decrease the thickness of the carbon paper, we focused herein on the concept of a Li-S cell model with a cathode comprising elemental sulfur and a 3D-carbon matrix (using this carbon paper as an example), leaving dimension optimization to a follow-up stage.

Testing conditions and discharge capacities of the cells are shown in Table 1. Figure 4 shows the initial charge/discharge capacities and voltage profiles of Li-S cells based on sulfur cathodes containing 70% sulfur (by weight) with different structure and total sulfur loading per cm^2^. The specific capacity per cm^2^ was calculated for fully discharged cathodes at a rate of 0.2 C. The specific energy density was calculated per the weight of the cathodes only. Using carbon-coated Al foil current collectors provided high specific capacity (test 1); however, increasing the sulfur loading in the cathodes from 1.3 to 1.7 mg/cm^2^ resulted in a drastic decrease in the rate capabilities of the cells (test 2). This implies that sulfur loading cannot be increased simply by using a carbon-coated Al foil.

Electrodes comprising carbon paper (3D) loaded with 1.7 mg/cm^2^ showed an improved 0.1 C rate discharge capacity, indicating that this 3D structured current collector enhances the electronic conductivity to elemental sulfur in the *z* direction (test 3). However, the 0.2 C rate discharge capacity is still lower compared to electrodes based on carbon-coated Al foil with 1.3 mg/cm^2^ sulfur loading. Carbon paper (3D) alone as a current collector cannot achieve a high rate due to its insufficient electronic conductivity.

In the next stage, the 3D and metal current collectors were combined by either mechanically binding the carbon to the Al foil or placing it above the foil. In the latter case (test 4) the cell showed an initial 0.1 C rate discharge capacity; however, further charging failed, and the cell was short-circuited. We assume that there was a severe shuttling effect, which did not allow for continued cycling. However, mechanically adhered cells (test 5) showed initially high specific capacity at the 0.1 C and 0.2 C rates after assembly.

Only in the case when the 3D and metal current collectors were adhered together was high capacity with both high S loading and high rate obtained. Therefore, in subsequent sections, bound carbon paper and carbon-coated Al foil is used as our standard current collector for sulfur cathodes.

The combination of carbon-coated Al foil current collector and carbon paper as discussed above showed that the two parts (Al CC and activated carbon matrix) must be well adhered. As this indicates that adhesion between these materials is important, different binders were studied. The adhesion was quantified by peeling tests. The test electrodes were adhered to the surface by a strong double-sided tape, and weight balls were placed in a cup suspended from the sulfur electrode that contained both carbon-coated Al foil and carbon paper. As the tape was strong, the location of the weakest adhesion was the interface between the two parts of the composite electrodes (the Al CC and the sulfur-impregnated activated carbon matrix) as confirmed by the peeling tests.

Focusing on swelling, PEO binder was chosen to prevent shuttling; however, it interacted with the electrolyte solution. As the carbon-coated Al foils and sulfur-impregnated activated carbon paper matrices were bound for enhancing electrode rate performance, this interface should have had strong adhesion. It seems that adhesion should weaken when the binder interacts with the electrolyte solution and swells. In this set of experiments, we changed the binders for the interface between the carbon-coated Al foil and the activated carbon paper matrix. PEO and PVDF were used as binders for the interface between these two parts of the composite electrodes. PVDF was selected, as it is known as a non-swelling binder. As shown in the previous section, PEO was used for the sulfur slurry in both cases. As it is desired to improve the adhesive stability while maintaining contact with the electrolyte solution, the adhesive effect of the two binders for the composite (bilayered) electrodes was tested with both dry electrodes and electrodes in contact with the electrolyte solutions.

Figure 5 shows the results of the peeling tests. Peeling intensity is shown on the *y* axis; thus, the interaction of the electrolyte solution (level of swelling) is calculated by measuring the weight and the electrode area. When PVDF was used as the binder, the electrodes showed strong adhesion. After the electrodes were placed into DOL/DME solution, the peeling intensity decreased in both electrodes with PEO and PVDF. However, adhesion of the latter was stronger even before placing into solution, suggesting a lower probability of losing the electronic conductivity by peeling during cycling.

Cycling tests with different binders for the interface between the carbon-coated Al foil and the sulfur-impregnated activated carbon paper matrix in the composite sulfur electrodes were carried out as shown in Figure 6. The sulfur loading was increased to 3.9 mg/cm^2^ to achieve practical areal capacity. The number of cycles was also increased to 150. Only the second binders (between carbon paper and carbon-coated Al foil) were changed. PEO was used as the first binder (for making S slurry) in both tests. As this work intends to demonstrate a new concept of composite sulfur cathodes rather than optimized electrodes, 150 cycles seemed quite sufficient for demonstrating feasibility.

PVDF in the interface between the two components in the composite cathodes decreased the initial gravimetric capacity. However, due to increased S loading, the initial areal capacity was still maintained above 2.0 mAh/cm^2^. At the initial stage of cycling, slight increase in capacity due to wetting was confirmed. After this, cells with PVDF showed significantly better cycling stability attributed to a strong adhesion between the two components of the composite cathodes, which improved the electronic conductivity of the electrodes. Figure 7 compares the charge/discharge voltage profiles of the electrodes containing different binders at initial 0.1 C and second-cycle 0.2 C rates in experiments with the same specific capacity over 150 cycles. At both rates the electrodes showed typical charge/discharge profiles of Li-S batteries. Cells with PVDF exhibited a lower capacity, however their voltage profiles were typical of Li-S cells. Interestingly, different charge/discharge voltage profiles were obtained for electrodes with the same initial specific capacity after 81 cycles (PEO) and 70 cycles (PVDF).

Usually, the voltage drop/increase at the beginning of charge/discharge cycling is attributed to the resistance of the electrodes; however, with Li-S cells we cannot discuss the electronic conductivity at the beginning of charge, as insulating Li_2_S precipitates on the cathode. Therefore, the first stage of the charging process relates to oxidation of an insulating material. However, we can still discuss the electronic conductivity of composite sulfur electrodes at the beginning of discharge. The electrodes with PEO showed a lower voltage plateau, reflecting an initial IR drop and consequently lower electronic conductivity compared to electrodes with PVDF binder. This result regarding the voltage profile at the beginning of discharge supports our peeling test. We conclude that the second cycling performance was improved by achieving better adhesion using PVDF between the two cathode components, resulting in more stable electronic conductivity.

## 3. Experimental Section

Elemental sulfur (99.98%, Sigma-Aldrich, Rehovot, Israel), carbon black (TIMCAL Super C65, Bodio, Switzerland), polyethylene-oxide and polyvinyl-pyrrolidone K90 (both from Sigma-Aldrich) at 70/20/9.5/0.5 weight ratio were mixed by ball milling with water as the solvent and coated on aluminum foil, a carbon-coated aluminum current collector, and carbon paper, with a sulfur loading of 1.3–3.9 mg/cm^2^. The weights of the 11 µm thick carbon-coated aluminum current collector and 100 µm thick carbon paper were 3.5 and 10.0 mg/cm^2^, respectively. Sulfur slurry–coated carbon papers were adhered to carbon-coated current collectors with 0.08 mg/cm^2^ of swelling PEO or PVDF binder; 10.5 cm^2^ electrodes were dried for 6 h at 60 °C and dried again for 3 h at 60 °C in vacuum prior to assembly.

Pouch cells were assembled with 125 μm thick Li foil anodes, Celgard 2500 polypropylene separators (Celgard, Charlotte, NC, USA), and 500 µL of 1 M LiTFSI DOL/DME electrolyte, with 0.1 M LiNO_3_ and 0.2 M Li_2_S_6_ additives added to suppress shuttling [10]. All sample preparations except the slurry making process were carried out in a glovebox (MBRAUN, Garching, Germany) under pure argon with less than 1 ppm of water. Electrochemical measurements were performed using computerized multichannel electrochemical measurement equipment from Arbin Instruments (College Station, TX, USA). Constant current cycling measurements were carried out to assess the stability of each system (electrodes, solutions). Cycling tests were carried out at 0.2 C rate (1 C = 1674 mAh/g) at 30 °C. Peeling tests were done using strong double-sided tape and weight balls.

## 4. Conclusions

This work suggests a new approach for the preparation of practical sulfur cathodes with high areal loading of sulfur that is suitable for mass production. Accordingly, new types of cathodes with elemental sulfur as the active mass were studied. Several important conclusions were reached:A suitable current collector and binder are important, even for the simplest electrodes. Starting with a very simple electrode configuration—a layer of sulfur spread with a binder on an Al foil current collector (while maintaining a high sulfur content of around 70% by weight)—an initial capacity approaching the maximal practical limit (around 1200 mAh/g) could be achieved at the 0.2 C rate. However, this could be reached only at a low sulfur loading of 1.3 mg/cm^2^. With higher loading, the insulating nature of sulfur does not allow reaching such high specific capacities. The carbon coating on the Al foil current collector and the use of a PEO binder that swells in the electrolyte solution are important. The swollen PEO encapsulates the sulfur, slowing the rapid dissolution of Li*_m_*S*_n_* reduction products from the cathode to the electrolyte solution.Composite electrodes based on sulfur impregnation into three-dimensional activated carbonaceous matrices are advantageous. Simple configurations based on the carbon-coated Al foil current collector limit pronouncedly effective sulfur loading. For instance, increasing areal loading from 1.3 to 1.5 mg/cm^2^ drastically reduced the obtainable specific capacity from around 820 to 430 mAh/g. Electrodes comprising 3D porous carbon matrices in which the sulfur is encapsulated enabled a drastic increase in areal loading to around 4 mg/cm^2^ and yet reached specific capacities that approach 800 mAh/g.Good adhesion should be maintained between the aluminum foil current collector and the sulfur-impregnated porous carbonaceous matrix that forms the 3D structure. Our favorite composite sulfur electrodes comprise two parts. The adhesion between them is important and requires selection of a special binder. PEO may serve as a good binder for the sulfur-containing matrices as its swelling encapsulates the sulfur active mass and mitigates detrimental interactions such as dissolution of Li*_m_*S*_n_* species from the cathode. However, due to swelling, it cannot serve as a good binder that avoids peeling of the active mass from the current collector during cycling. Polyvinylidene fluoride, a non-swelling binder, provided reasonable cycling stability with high sulfur loading.

Three factors enable improvement of the performance of elemental sulfur cathodes containing 70% of active mass and high areal loading in terms of high specific capacity, reasonable rate capability, and cycling stability without exotic structures and/or electrolyte solutions. However, this demonstration, resulting from a stepwise systematic study, is far from practical. Further optimization may include a judicious use of additives in solution, successful anode pretreatments by protective surface reactions, and fine adjustment of the cathode components while maintaining high sulfur and areal loading.

It is important to emphasize several other important points that are connected to development of sulfur electrodes in general and to this work in particular. There are many publications on Li-S cells demonstrating ‘exotic’ electrode structures containing complex composite structures addressing this battery technology for electric vehicles (hence, aiming for very prolonged cycle life of Li-S cells through the complex electrode design). The major advantage of Li-S cells is the possibility to reach high energy density despite the relatively low operational potential (around 2 V compared to 3.8 V in average for Li ion batteries). The high energy density may be reached owing to the high theoretical specific capacity of the Li–S reaction to form Li_2_S as the final product (1675 mAh/g, ~6× greater than the highest theoretical capacity of Li ion insertion cathodes).

In practice, sulfur cathodes can reach, in most types of composite electrodes, a specific capacity <1000 mAh/g sulfur. Thereby, a key issue in developing practical Li-S batteries is cathodes in which specific loading of sulfur per unit area is the highest possible. Such electrodes should always suffer from limited cycle life. However, the practical future of this technology is to maximize specific capacity of the sulfur cathodes (and, hence, to maximize the specific energy of the full Li-S cells) on the account of cycle life. This means that Li-S batteries are not good candidates for powering EVs. However, optimizing the high specific capacity of sulfur cathodes in Li-S cells and maximizing cycle life even to a few hundred cycles can bring these systems to practical horizons where the high energy density is a key factor (such as the field of unmanned drones). In the work behind this paper and follow-up stages, we aim to develop practical and scalable high specific capacity sulfur cathodes with high areal loading that will be able to undergo a sufficient number (hundreds) of cycles with reasonably stable capacity.

## Figures and Tables

**Figure 1 molecules-28-04568-f001:**
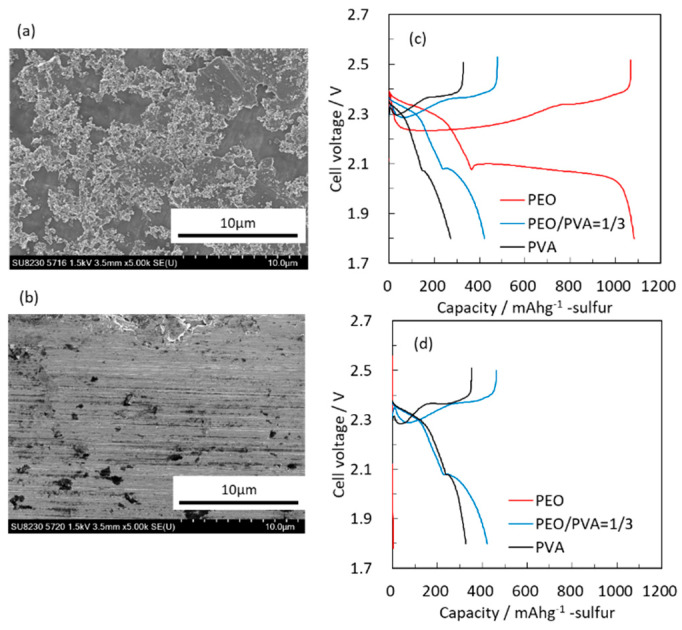
SEM images of carbon-coated (**a**) and glossy (**b**) Al current collectors and initial 0.1 C and 0.2 C charge/discharge voltage profiles in 1 M LiTFSI in 1:1 DOL/DME (**c**,**d**), respectively.

**Figure 2 molecules-28-04568-f002:**
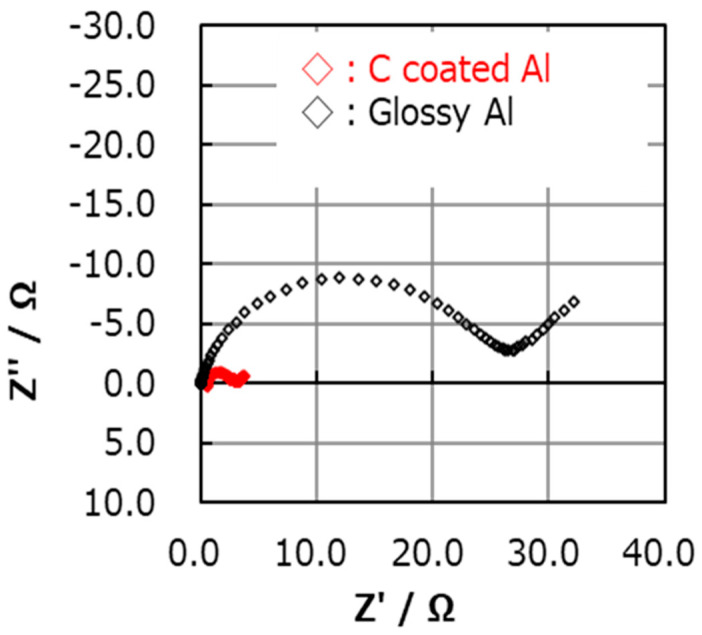
Initial impedance spectra of composite sulfur electrodes (presented as Nyquist plots from 0.1 Hz to 1 MHz) with glossy and carbon-coated current collectors. 7/2/1 S/super P/ PEO component with 1.3 mg/cm^2^ sulfur loading and 1 M LiTFSI in 1:1 DOL/DME.

**Figure 3 molecules-28-04568-f003:**
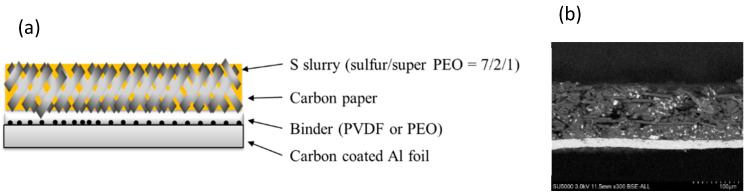
(**a**) Schematic image of the sulfur cathode with carbon-coated Al foil and carbon paper. (**b**) Cross-sectional SEM image of the electrode with S loading of 1.7 mg/cm^2^.

**Figure 4 molecules-28-04568-f004:**
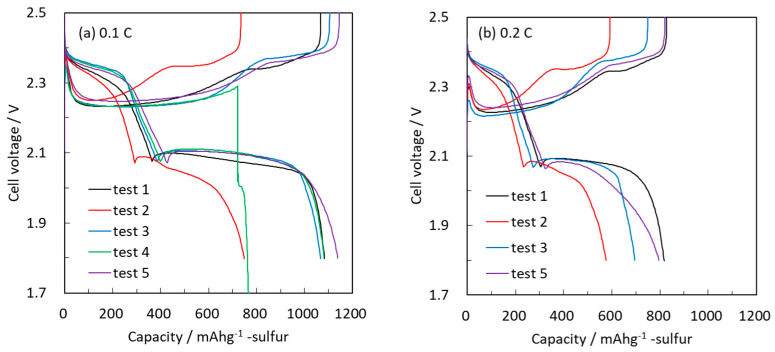
Initial charge/discharge voltage profiles of Li-S cells with cathodes containing 70% elemental sulfur with different S loading and current collectors: carbon-coated Al with a loading of (test 1) 1.3 and (test 2) 1.7 mg/cm^2^, (test 3) carbon paper with a loading of 1.7 mg/cm^2^, (test 4) carbon paper (without adherence) with 1.7 mg/cm^2^ S, and (test 5) well-adhered carbon paper loaded with 1.7 mg/cm^2^ S. All solutions contained 1 M LiTFSI and 0.2 M Li-nitrate in 1:1 DOL/DME.

**Figure 5 molecules-28-04568-f005:**
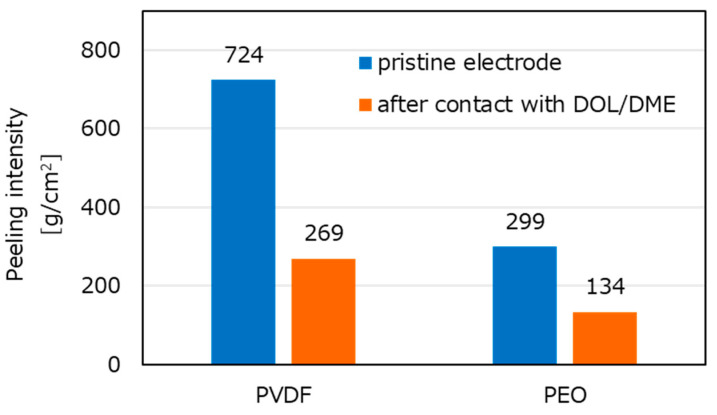
Results of peeling tests with PVDF and PEO binders for the interface between the carbon-coated Al foil and activated carbon paper before and after contact with the electrolyte solution.

**Figure 6 molecules-28-04568-f006:**
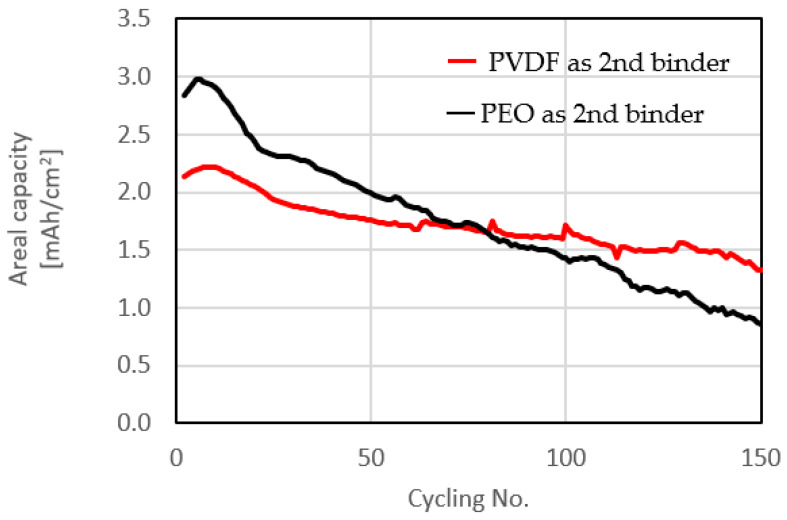
Cycling performance of cells comprising cathodes with high sulfur loading with different binders for the interface between the two components of the composite cathode [4].

**Figure 7 molecules-28-04568-f007:**
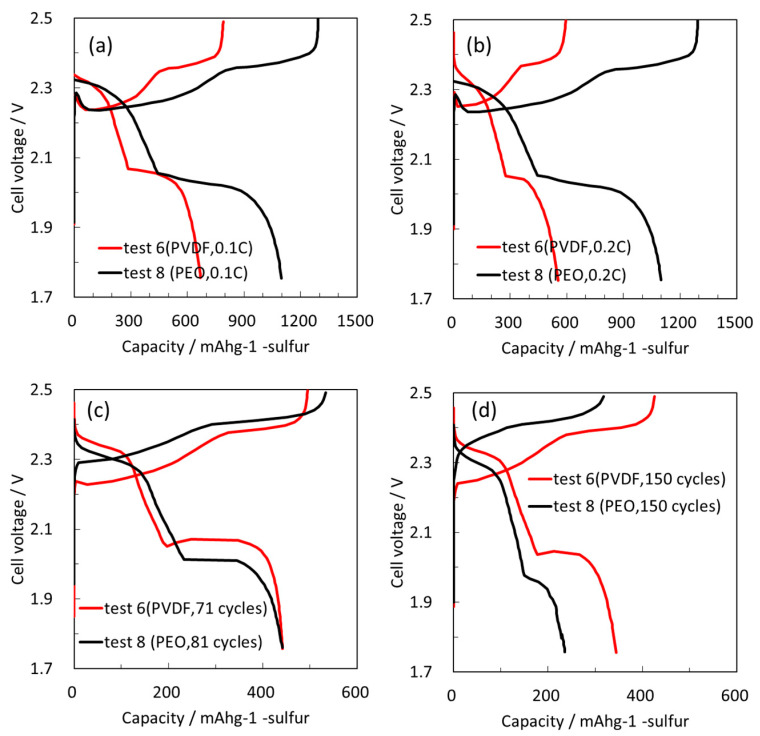
Comparison of charge/discharge voltage profiles with PEO and PVDF binders in galvanostatic cycling in standard electrolyte solutions at initial (**a**), second (**b**), 70th and 81st (same specific capacity) (**c**), and 150th (**d**) cycles. The first cycle was at 0.1 C, with further cycling at 0.2 C.

**Table 1 molecules-28-04568-t001:** Testing conditions of Li-S batteries with different current collectors and S loading.

Name	Current Collector	S Loading (mAh/cm^2^)	0.1 C Discharge (mAh/g)	0.2 C Discharge (mAh/g)
test 1	Carbon coated Al	1.3	1084	817
test 2	Carbon coated Al	1.7	608	431
test 3	Carbon paper	1.7	1067	695
test 4	C-coated Al and carbon paper (without adherence)	1.7	1082	0
test 5	C-coated Al and well adhered carbon paper	1.7	1139	796

## Data Availability

The data presented in this study is available upon request from the corresponding author.

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
