# Peer review of "Stable High-Capacity Elemental Sulfur Cathodes with Simple Process for Lithium Sulfur Batteries"

_molecules, 2023, doi:10.3390/molecules28124568_

Round 1

Reviewer 1 Report

 Sawada et al. used the selected binders and carbonaceous matrices impregnated with the active mass was used to enhance the performance of sulfur-based cathodes. They have obtained the maximum specific capacity of 805 mAh/g 20 (areal capacity of 2.2 mAh/cm2). The manuscript is suggested to be accepted after the following issues are addressed.

1)      Many spelling, grammatical, units and typo errors are present in this paper, and the authors should double-check and revise them thoroughly.

2)      Most of the abstract part is related to the general information about materials, and the authors should revise the abstract of the manuscript by adding the main outcomes.

3)      The authors use the carbon-coated aluminium current collector and check the effect of two binders on the bases of adhesion. What about the adhesion of these binders with other collector currents?

4)      There is no structure or surface information of materials or electrodes. The authors should provide the information.

5)        There is no information about the weight of the three materials.

6)      The authors measure up to 150 cycles to check the small device's stability for practical applications. The authors should measure higher order

Author Response

Many spelling, grammatical, units and typo errors are present in this paper, and the authors should double-check and revise them thoroughly. 

Many thanks for this comment. We proofread the revised paper and carefully checked the language and edited further as needed.

Most of the abstract part is related to the general information about materials, and the authors should revise the abstract of the manuscript by adding the main outcomes.

Many thanks. A main outcome is a judicious selection of current collectors and binders. We added to the abstract notes on selection of suitable components for composite sulfur electrodes (please see highlighted text).

The authors use the carbon-coated aluminium current collector and check the effect of two binders on the bases of adhesion. What about the adhesion of these binders with other current collectors ?

SUS304 and scratched Al foils were tested for comparison. Scratched Al foil improved adhesion, and some improvement in performance was confirmed. We thus concluded that the physical contact between the carbon and the metal is important. We did not see a need to check other current collectors.

There is no information about the weight of the three materials.

Many thanks for pointing out this omission. We added details on the weight of the materials. Please note that the goal was to demonstrate a concept for composite sulfur cathodes that may allow high loading. We do not show results of quantitative analysis, but rather provide a qualitative demonstration of the proof of concept. A discussion on relative weights should be conducted after concluding an optimization procedure. This is now explained.

The authors measure up to 150 cycles to check the small device's stability for practical applications. The authors should measure higher order

We thank the reviewer for this suggestion. As explained above, the goal was to show a new idea of sulfur cathode design that allows high loading without involving exotic structures and/or electrolyte solutions. Reviewing the literature, 150 cycles with elemental sulfur at loading >2 mAh/cm2 per cell seems quite reasonable. We did not intend to complete the optimization, but rather to demonstrate the new concept. Therefore, prolonged cycling experiments are not included. This is explained in the revised manuscript.

Reviewer 2 Report

This article deals with understanding the influence of current collectors, binders, and 3D carbonaceous matrices. The authors have varied various factors to achieve high active material loading, which has shown reasonable rate capability and cycling stability. However, the representation of the work is inferior in terms of figure arrangements and scientific explanation. A few of my suggestions are given below;

1.      Figure quality should be improved. The scale bar is barely visible in the SEM micrographs (Fig. 1). The Axis should have adjected in the scale of 500 and multiples. (C in capacity should be capital). The cycle number should be plotted separately, as it is complicated to distinguish the charge/discharge profile.

2.      What is the loading of carbon on the Al foil? Does the thickness of carbon coating affect the electrochemical performance?

3.      It is challenging to distinguish the charge-discharge profile in all the figures (Figure 1 and Figure 4). The cycle number should be represented with a different color to distinguish it properly. The axis description is missing in Figure 4.  

4.      Impedance graph is missing an equivalent circuit. The authors are suggested to include the equivalent circuit.

5.      There is no proper explanation for the gradual increase in the capacity (First few cycles) followed by a rapid decrease of reversible capacity in the case of PEO based binder in Figure 7.

6.      English should be carefully worked. It should be vastly improved, and the manuscript should be proofread.

7.      Many minor mistakes need to be corrected

“(2-c). Using unique….” Should be “(2-c): Using unique….”

“(Figure 1.) shows a SEM image….” Should be “Figure 1. shows a SEM image…..”

Author Response

Figure quality should be improved. The scale bar is barely visible in the SEM micrographs (Fig. 1). The Axis should have adjected in the scale of 500 and multiples. (C in capacity should be capital). The cycle number should be plotted separately, as it is complicated to distinguish the charge/discharge profile.

Many thanks for the comment. We modified the figures carefully. The scale of 500 and multiples was not applied as it does not appear to have a significant contribution.

What is the loading of carbon on the Al foil? Does the thickness of carbon coating affect the electrochemical performance?

A commercial carbon coated Al foil was used with 11 µm thickness. The thickness of the carbon coating was less than 1 µm. We have no data about the influence of the coating thickness on the electrochemical performance, but we assume it may have some effect. This issue relates to follow-up optimization efforts.

It is challenging to distinguish the charge-discharge profile in all the figures (Figure 1 and Figure 4). The cycle number should be represented with a different color to distinguish it properly. The axis description is missing in Figure 4. 

Many thanks for pointing this out. The figures were modified as suggested.

Impedance graph is missing an equivalent circuit. The authors are suggested to include the equivalent circuit.

Thank you for this suggestion. We are very experienced in EIS analysis of various batteries. For Li-S cells containing composite sulfur electrodes, the importance of EIS measurements is only qualitative, allowing a rough comparison among systems, serving also as an in-situ spectroscopic tool to follow general changes during cycling. EIS measurements are good for following changes and controlling stability, however EIS of these composite systems reflects many different time-constants working in parallel. Therefore, simulation of spectra by equivalent circuits is meaningless (many circuits may fit the same spectrum).

There is no proper explanation for the gradual increase in the capacity (First few cycles) followed by a rapid decrease of reversible capacity in the case of PEO based binder in Figure 7.

The increase in capacity at the beginning is due to advancement of wetting. At the beginning of the cycling, wetting may be incomplete. Obviously, full wetting is reached during cycling. This is now explained (see p. 7).

English should be carefully worked. It should be vastly improved, and the manuscript should be proofread.

Many thanks. The language was carefully revised and the manuscript was thoroughly proofread.

Reviewer 3 Report

The manuscript (Stable High Capacity Elemental Sulfur Cathodes with Facile Process for Lithium Sulfur Batteries) presented composite current collectors with optimizing of the binders and carbon materials. However, several issues recued the importance of this paper. The main issues were the insufficient data and poor performance and analysis.

- In general, the manuscript was rough. Different units were randomly used. Several units were wrong in the writing and in the plots. The data were insufficiently analyzed and mixed together without considering the effects of different cells. The figures are roughly prepared.

- No material analysis. The electrochemical testing and analysis were rough analyzed and put together without organized.

- The electrochemical testing and cell analysis were mostly used low S loading. 1.3-1.7 mg / cm2 caused a huge error in the cell testing.

- According to the data presentation, the replacement of binder did not improve the performance. The cell either showed low capacity or high capacity fade.

- You should report your cells’ sulfur content when you highlight low S content as an issue.

- You should report the electrolyte volume.

- You had a lot of cells in the data. But, the useful data showed a different property and low performance than those cells in the analysis.

- You should update the reference so that will know the current LSB research.

- The carbon paper used was heavy and thick. The S content was low.

- The added Li2S6 should be reported with the amount and the capacity given.

- the cell information should not be in figures. The information should be clearly reported in the experimental section. Additionally, the cell information in plots were randomly presented.

- figure 7 had wrong S loading reported in figure and in the caption.

- In all, the experiments, data, and discussion were not organized. The paper was rough and need proofreading. Currently, it was suggested to be rejected.

Author Response

- In general, the manuscript was rough. Different units were randomly used. Several units were wrong in the writing and in the plots. The data were insufficiently analyzed and mixed together without considering the effects of different cells. The figures are roughly prepared.

Thanks. The manuscript was smoothened accordingly.

- No material analysis. The electrochemical testing and analysis were rough analyzed and put together without organized.

Surface analysis of Li electrodes is not required as it is well known for the electrochemical systems used here (based on our previous successful work) . Chemical analysis of sulfur cathodes is also not needed as we know how sulfur reacts and which products are formed. What is important indeed is the morphological analysis. We provide morphological data, explaining the concept that we wish to promote.

- The electrochemical testing and cell analysis were mostly used low S loading. 1.3-1.7 mg / cm2 caused a huge error in the cell testing.

We used low S loading to verify our concept. The loading was increased from 1.3 to 1.7 mg/cm2 as it helped us to understand the trends in performance and the need for a 3D structure for increasing loading. Indeed, we could increase the S loading up to 3.9 mg/cm2 by employing electrodes with 3D structure to improve cell capacity.

- According to the data presentation, the replacement of binder did not improve the performance. The cell either showed low capacity or pronounced capacity fading.

The performance improved using PEO and PVDF rather than PEO alone due to strong adhesion as shown in Figure 6 and described on p. 7.

- You should report your cells’ sulfur content when you highlight low S content as an issue.

We mention the S content as % in a mixture of S/carbon/binder in the composite cathodes, rather than in terms of % in the cells. The idea of this paper was to offer a new concept of composite sulfur cathodes that may allow high sulfur loading. The electrodes and cells are not optimal. Further engineering is required to complete optimization and present full cells with commercial importance. Further work (under way) includes modifying the thickness of the current corrector, minimizing the sulfur/electrolyte solution ratio, optimizing electrode porosity, and using the thinnest Li anodes possible. Only then does it make sense to report the percentage of active sulfur in the cells. It should be emphasized that such optimization is beyond the scope of the paper and cannot be completed properly in a university setting. We present here cell components that can be considered as a starting point for optimization. This is discussed in the revised manuscript (see p. 9).

- You should report the electrolyte volume. 

The electrolyte volume was added in the experimental section. Please note that the E/S ratio is critically important for evaluating practical cells. Our cells are not optimized in terms of high specific capacity/energy per unit weight and volume. Naturally our laboratory cells contain more free space than practical cells produced by usual industrial processes. Therefore, the E/S ratio is not important here.

- You had a lot of cells in the data. But, the useful data showed a different property and low performance than those cells in the analysis.

Thank you for this comment, however we do not fully understand it. We performed indeed many experiments but show average results of selected cells that in our opinion are representative. We refined the data again in the framework of our revision and made sure that the presented results are coherent.

- You should update the reference so that will know the current LSB research.

This research focused on the possibility to prepare and test relatively simple electrodes with high loading of elemental S as the active material in Li-S cathodes without using any sophisticate/expensive materials and preparation methods. The work behind the paper is a first stage before industrial development. This is now clarified and emphasized.

- The carbon paper used was heavy and thick. The S content was low.

Many thanks for this important comment. We fully agree with the reviewer that the electrodes and cells presented here are far from optimal. We suggest a new type of composite sulfur electrodes and demonstrate selected components that can be considered as a reasonable starting point. We show relevant data for that. A follow-up work is optimization – minimizing the specific weight of all non-active components (except sulfur). This is explained in the revised manuscript.

 - The added Li2S6 should be reported with the amount and the capacity given.

Li2S6 is not used here as a component that adds specific capacity, but rather as an additive to the electrolyte solution for making stable surface films on the Li anodes (well-known from our previous studies). Therefore, the amount of added Li2S6 is much smaller than that contained in cells based on the sulfur catholyte concept. We tested cells with and without Li2S6. The initial discharge capacity showed no difference. This additive was important to increase the cycle life (well-known owing to the effect on Li anode passivation).

- the cell information should not be in figures. The information should be clearly reported in the experimental section. Additionally, the cell information in plots were randomly presented.

Many thanks for this useful suggestion. We modified the figures and the experimental section accordingly.

- figure 7 had wrong S loading reported in figure and in the caption.

Thanks. Figure 7 was removed (the caption was fine).

Round 2

Reviewer 1 Report

Accept in present form.

Author Response

Many thanks to reviewer 1 and his/her recommendation to accept the revised paper in its present form.

Reviewer 2 Report

Authors have addressed the issues 

Author Response

Many thanks to reviewer 2 and his/her recommendation to accept the revised paper in its present form.

Reviewer 3 Report

- The authors reject most of the request revision.  Although wordy information is added, the content is still not suitable for publication.

- The requested material and electrochemical analysis are necessary for this work.  You need data for this manuscript, not comments from your previous work.

- As the S-load increases, the cell properties decrease.  The analysis is wrong from this work to practical batteries.

- E/S ratio is necessary to be reported and important here.  And it is necessary information for publication in many journals.  Why do you comment "Therefore, the E/S ratio is not important here" in your paper?

- The coulombic efficiency is still missing.

- If you plan to do no modification by the excuse of verifying a concept. A criterion can be your reference.  To verify your Li/S concept, the areal capacity should be over 4 mAh/g for getting reliable data for analysis.

- This resubmission shows almost no improvement, and the authors refuse to make any changes.  So I have to recommend rejection this time.

Author Response

Our Response:

It is not true that we rejected any requested revision. Please see below our response to the requests of reviewer 3 and the change made in our revision (with which reviewer 3 is still not happy).

The revised manuscript contains enough materials for a full paper, as reviewers 1 & 2 agreed. What reviewer 3 wants from us is to carry out a lot of experimental work, to add new data, to change loading, to calculate E/S ratio before optimization (useless!) or to complete optimization to provide useful E/S rations. However, to complete optimization means to work one year more. This is not a fair request.

Of course, if we work more, we can add more results, but this is true for any paper.

Our opinion (confirmed by reviewers 1 & 2) that the revised paper contains enough data for a single article. We say in that paper that the work is not completed yet and this paper and the work behind it provide a strong incentive to continue toward full practical cells, using the same ideas described herein.

Please see below and realize that we answered ALL the comments of reviewer 3 from the first reviewing round. His/her claim that we rejected his/her comments are not true.

Here are the comments of reviewer 3 from the first reviewing round (in black) and our response in red (including explanations how we revised the paper).

<Reviewer 3>

- In general, the manuscript was rough. Different units were randomly used. Several units were wrong in the writing and in the plots. The data were insufficiently analyzed and mixed together without considering the effects of different cells. The figures are roughly prepared.

→ Thanks. The manuscript was smoothened accordingly.

- No material analysis. The electrochemical testing and analysis were rough analyzed and put together without organized.

Surface analysis of Li electrodes is not required as it is well known for the electrochemical systems used here (based on our previous successful work) . Chemical analysis of sulfur cathodes is also not needed as we know how sulfur reacts and which products are formed. What is important indeed is the morphological analysis. We provide morphological data, explaining the concept that we wish to promote.

- The electrochemical testing and cell analysis were mostly used low S loading. 1.3-1.7 mg / cm2 caused a huge error in the cell testing.

We used low S loading to verify our concept. The loading was increased from 1.3 to 1.7 mg/cm2 as it helped us to understand the trends in performance and the need for a 3D structure for increasing loading. Indeed, we could increase the S loading up to 3.9 mg/cm2 by employing electrodes with 3D structure to improve cell capacity.

- According to the data presentation, the replacement of binder did not improve the performance. The cell either showed low capacity or pronounced capacity fading.

The performance was improved using PEO and PVDF rather than PEO alone due to strong adhesion as shown in Figure 6 and described on p. 7.

- You should report your cells’ sulfur content when you highlight low S content as an issue.

We mention the S content as % in a mixture of S/carbon/binder in the composite cathodes, rather than in terms of % in the cells. The idea of this paper was to offer a new concept of composite sulfur cathodes that may allow high sulfur loading. The electrodes and cells are not optimal. Further engineering is required to complete optimization and present full cells with commercial importance. Further work (under way) includes modifying the thickness of the current corrector, minimizing the sulfur/electrolyte solution ratio, optimizing electrode porosity, and using the thinnest Li anodes possible. Only then does it make sense to report the percentage of active sulfur in the cells. It should be emphasized that such optimization is beyond the scope of the paper and cannot be completed properly in a university setting. We present here cell components that can be considered as a starting point for optimization. This is discussed in the revised manuscript (see p. 9).

- You should report the electrolyte volume. 

The electrolyte volume was added in the experimental section. Please note that the E/S ratio is critically important for evaluating practical cells. Our cells are not optimized in terms of high specific capacity/energy per unit weight and volume. Naturally our laboratory cells contain more free space than practical cells produced by usual industrial processes. Therefore, the E/S ratio is not important here.

- You had a lot of cells in the data. But, the useful data showed a different property and low performance than those cells in the analysis.

Thank you for this comment, however we do not fully understand it. We performed indeed many experiments but show average results of selected cells that in our opinion are representative. We refined the data again in the framework of our revision and made sure that the presented results are coherent.

- You should update the reference so that will know the current LSB research.

This research focused on the possibility to prepare and test relatively simple electrodes with high loading of elemental S as the active material in Li-S cathodes without using any sophisticate/expensive materials and preparation methods. The work behind the paper is a first stage before industrial development. This is now clarified and emphasized.

- The carbon paper used was heavy and thick. The S content was low.

Many thanks for this important comment. We fully agree with the reviewer that the electrodes and cells presented here are far from optimal. We suggest a new type of composite sulfur electrodes and demonstrate selected components that can be considered as a reasonable starting point. We show relevant data for that. A follow-up work is optimization – minimizing the specific weight of all non-active components (except sulfur). This is explained in the revised manuscript.

 - The added Li2S6 should be reported with the amount and the capacity given.

Li2S6 is not used here as a component that adds specific capacity, but rather as an additive to the electrolyte solution for making stable surface films on the Li anodes (well-known from our previous studies). Therefore, the amount of added Li2S6 is much smaller than that contained in cells based on the sulfur catholyte concept. We tested cells with and without Li2S6. The initial discharge capacity showed no difference. This additive was important to increase the cycle life (well-known owing to the effect on Li anode passivation).

- the cell information should not be in figures. The information should be clearly reported in the experimental section. Additionally, the cell information in plots were randomly presented.

Many thanks for this useful suggestion. We modified the figures and the experimental section accordingly.

- figure 7 had wrong S loading reported in figure and in the caption.

Thanks. Figure 7 was removed (the caption was fine).